# Infectious Pneumonia and Lower Airway Microorganisms in Patients with Rheumatoid Arthritis

**DOI:** 10.3390/jcm10163552

**Published:** 2021-08-12

**Authors:** Shuhei Ideguchi, Kazuko Yamamoto, Masahiro Tahara, Tomohiro Koga, Shotaro Ide, Tatsuro Hirayama, Takahiro Takazono, Yoshifumi Imamura, Taiga Miyazaki, Noriho Sakamoto, Shimpei Morimoto, Koichi Izumikawa, Katsunori Yanagihara, Kazuto Ashizawa, Takatoshi Aoki, Atsushi Kawakami, Kazuhiro Yatera, Hiroshi Mukae

**Affiliations:** 1Department of Respiratory Medicine, Nagasaki University Graduate School of Biomedical Sciences, Nagasaki 852-8521, Japan; ideguchi.shuhei.pd@mail.hosp.go.jp (S.I.); hmukae@nagasaki-u.ac.jp (H.M.); 2Department of Respiratory Medicine, Nagasaki University Hospital, Nagasaki 852-8102, Japan; tatsuro_h_20@nagasaki-u.ac.jp (T.H.); takahiro-takazono@nagasaki-u.ac.jp (T.T.); yimamura@nagasaki-u.ac.jp (Y.I.); taiga-m@nagasaki-u.ac.jp (T.M.); nsakamot@nagasaki-u.ac.jp (N.S.); 3Department of Infection Control and Education Center, Nagasaki University Hospital, Nagasaki 852-8102, Japan; koizumik@nagasaki-u.ac.jp; 4Clinical Research Center, National Hospital Organization Nagasaki Medical Center, Omura 856-0835, Japan; 5Department of Respiratory Medicine, University of Occupational and Environmental Health Japan, Kitakyushu 807-8555, Japan; tahara-masahiro@med.uoeh-u.ac.jp (M.T.); yatera@med.uoeh-u.ac.jp (K.Y.); 6Department of Immunology and Rheumatology, Division of Advanced Preventive Medical Sciences, Nagasaki University Graduate School of Biomedical Sciences, Nagasaki 852-8521, Japan; tkoga@nagasaki-u.ac.jp (T.K.); atsushik@nagasaki-u.ac.jp (A.K.); 7Department of Respiratory Medicine, Isahaya General Hospital, Isahaya 854-8501, Japan; side.ngs@gmail.com; 8Innovation Platform & Office for Precision Medicine, Nagasaki University Graduate School of Biomedical Sciences, Nagasaki 852-8521, Japan; morimoto.s@nagasaki-u.ac.jp; 9Department of Laboratory Medicine, Nagasaki University Hospital, Nagasaki 852-8102, Japan; k-yanagi@nagasaki-u.ac.jp; 10Department of Clinical Oncology, Nagasaki University Graduate School of Biomedical Sciences, Nagasaki 852-8521, Japan; ashi@nagasaki-u.ac.jp; 11Department of Radiology, University of Occupational and Environmental Health Japan, Kitakyushu 807-8555, Japan; a-taka@med.uoeh-u.ac.jp

**Keywords:** rheumatoid arthritis, pneumonia, *Pseudomonas aeruginosa*

## Abstract

The relationship between microorganisms present in the lower respiratory tract and the subsequent incidence of pneumonia in patients with rheumatoid arthritis is unclear. A retrospective cohort study was designed to include a total of 121 patients with rheumatoid arthritis who underwent bronchoscopy at three hospitals between January 2008 and December 2017. Data on patient characteristics, microorganisms detected by bronchoscopy, and subsequent incidences of pneumonia were obtained from electronic medical records. Patients were divided into groups based on the microorganisms isolated from the lower respiratory tract. The cumulative incidence of pneumonia was assessed using the Kaplan–Meier method, and decision tree analysis was performed to analyze the relation between the presence of microorganisms and the occurrence of pneumonia. The most frequently isolated microbes were *Pseudomonas aeruginosa*, *Staphylococcus aureus*, and *Haemophilus influenzae*. Patients whose samples tested negative for bacteria or positive for normal oral flora were included in the control group. The rate of the subsequent incidence of pneumonia was higher in the *P. aeruginosa* group than in the control group (*p* = 0.026), and decision tree analysis suggested that *P. aeruginosa* and patient performance status were two important factors for predicting the incidence of pneumonia. In patients with rheumatoid arthritis, the presence of *P. aeruginosa* in the lower respiratory tract was associated with the subsequent incidence of pneumonia.

## 1. Introduction

The high incidence of pneumonia and associated mortality in patients with rheumatoid arthritis (RA) is a critical concern [1]. Airway diseases (ADs) such as bronchiectasis are common complications of RA, with a rate of 10–30% [2,3]; patients with RA and ADs therefore have significantly poor prognosis [4]. *Pseudomonas aeruginosa* increases the risk of mortality and exacerbation in patients with bronchiectasis [4,5]; however, the relationship between airway microorganisms and the incidence of pneumonia in patients with RA has not yet been clarified. Thus, we aimed to identify whether the presence of microorganisms in the lower respiratory tract (LRT), particularly *P. aeruginosa*, correlated with the occurrence of pneumonia in patients with RA.

## 2. Materials and Methods

### 2.1. Patients and Study Design

This retrospective cohort study was conducted at the Nagasaki University Hospital, University of Occupational and Environmental Health Japan, and Isahaya General Hospital, between January 2008 and December 2017. This multicenter study was conducted in compliance with the Declaration of Helsinki and was approved by the Ethics Committees of the participating institutions (Nagasaki University Hospital, Approval Number: 19061714; University of Occupational and Environmental Health Japan, Approval Number: 19-044; Isahaya General Hospital, Approval Number: 2020 9). The requirement for patient consent was waived due to the retrospective nature of the study, which ensured anonymity.

The inclusion criteria were as follows: age ≥ 20 years, an RA diagnosis by rheumatologists certified by the Japan College of Rheumatology [6], and bronchoscopy performed by a pulmonary physician. We excluded patients if they had active infections requiring antimicrobial agents at the time of bronchoscopy; malignancy diagnosed within 5 years of the study; tracheotomy; cystic fibrosis (CF); or at least one of the following conditions: human immunodeficiency virus infection, a CD4 cell count of <350/μL, solid organ transplantation, or neutropenia of <500 cells/μL. Patient characteristics and RA status were extracted from the hospitals’ electronic medical records.

### 2.2. Evaluation of Bronchiectasis

Chest computed tomography (CT) was performed before bronchoscopy. Bronchiectasis was assessed by blinded reads, based on the modified Reiff score [4], by two board-certified radiologists from the Japanese Radiological Society with an average of 25 years of experience. The Reiff score indicates the degree of bronchial dilatation (tubular, 1; varicose, 2; and cystic, 3) and number of lobes involved. The lingular segment was evaluated as an independent lobe. Scores ranged from 1 to 18, and patients without bronchodilation were assigned a score of 0.

### 2.3. Identification of the Lower-Respiratory-Tract-Colonizing Microbes

To detect microorganisms in the LRT, bronchial lavage or intratracheal sputum collected by bronchoscopy was directly cultured using blood with chocolate agar (37 °C, 5% CO_2_), anero Columbia agar (anaerobic chamber), and bromothymol blue with chromagar (37 °C) for detecting Gram-positive, Gram-negative, and anaerobic bacteria and fungi.

### 2.4. Endpoint

The data were censored on 31 March 2019. Pneumonia incidence and pneumonia-free survival, calculated from the date of bronchoscopy to the date of pneumonia diagnosis, were recorded. Patients who did not develop pneumonia or were lost to follow-up were censored at the date of last contact. Pneumonia was confirmed by the presence of new lung infiltrates and at least one of the following acute respiratory symptoms: cough, sputum production, dyspnea, a body temperature of ≥38.0 °C, abnormal auscultatory findings, and leukocyte counts of >10,000 cells/μL or <4000 cells/μL. The incidence of pneumonia was identified from electronic medical records, which also provided data regarding antimicrobial use history.

### 2.5. Statistical Analysis

The cumulative incidence of pneumonia was estimated using the Kaplan–Meier method and log-rank test. A *p*-value of <0.05 was considered statistically significant. Recursive partitioning was performed by the CART algorithm in R [7,8]. Statistical analyses were performed using GraphPad Prism 5 (GraphPad Software, San Diego, CA, USA), JMP^®^ 13 (SAS Institute, Cary, NC, USA) and the R environment (version 4.1.1, R Foundation, Vienna, Austria) [9].

## 3. Results

### 3.1. Eligible Patients

Among the 228 patients with RA who underwent bronchoscopy during the study period, 107 were excluded from the study, including 71 with active infections (pneumonia, 43; non-tuberculosis mycobacteria, 16; cryptococcosis, 7; aspergillosis, 3; pneumocystis, 1; and other mycosis, 1), 33 with malignancies (lung, 28; renal, esophageal, ovarian, endometrial, and adult T-cell leukemia-lymphoma, 1 each), 1 with a tracheotomy, and 1 with solid organ transplantation. No data were available for 1 patient who was also excluded. Consequently, 121 patients were included in the study. Patients with RA mainly underwent bronchoscopy to identify the presence of infections before or during treatment for RA; this included further examination for mycobacteria.

### 3.2. Lower Respiratory Tract Microorganisms

The microbes detected in the LRT samples are listed in Figure 1. The LRT samples tested positive in 100 patients (82.6%), including 41 (33.9%) with normal oral flora. Among the frequently isolated pathogens, *P. aeruginosa* (13.2%), *Staphylococcus aureus* (12.4%), and *Haemophilus influenzae* (5.0%) were the most prominent. The other pathogens were *Streptococcus pneumoniae* (1.7%), other streptococci (3.3%), other Gram-negative bacilli (5.0%), anaerobic bacteria (1.7%), non-tuberculous mycobacteria (4.1%), Candida species (3.3%), and other fungi (2.5%). Multiple pathogens were detected simultaneously in four patients.

### 3.3. Patient Groups Based on Lower Respiratory Tract Microbes

Patients with RA were divided into groups depending on the microorganisms isolated from the LRT, namely, *P. aeruginosa* (Pa group), *S. aureus* (Sa group), and *H. influenzae* (Hi group). The patients whose samples tested negative for bacteria (*n* = 21) or positive for normal oral flora (*n* = 41) were included in the control group and were compared with the abovementioned bacteria groups (Table 1). The other lung-specific pathogens, such as *S. pneumoniae*, were excluded from this analysis.

### 3.4. Patient Backgrounds per Group

Age, sex, smoking history, RA duration, and observation period did not differ between the groups. The female dominance in the study population reflected the patterns observed in Japanese patients with RA [10]. The Disease Activity Score-28 for RA with erythrocyte sedimentation rate (DAS28-ESR), one of the indexes for evaluating the disease activity of RA based on the ESR and number of tender and swollen joints, was relatively high in the control group. The incidences of diabetes mellitus (DM), chronic obstructive pulmonary disease (COPD), and interstitial lung disease (ILD) were relatively low in the Sa group. The rate of glucocorticoid (GC) usage in the Sa group was low. Long-term macrolides (≥1 month) were more commonly administered in the Pa group. The Pa and Sa groups showed a higher degree of bronchiectasis, detected by the modified Reiff score.

### 3.5. Incidence of Pneumonia

The annual incidence rates of pneumonia per 1000 patients were 100, 62, 132, and 33, in the Pa, Sa, Hi, and control groups, respectively. The Kaplan–Meier curves are shown in Figure 2. Log-rank tests revealed that the Pa group had a significantly higher incidence of pneumonia (*p* = 0.026).

### 3.6. Decision Tree Analysis

The DAS-28 score was missing in many cases and was therefore excluded from the analysis. There were three and one missing values for the duration of GC use and RA, respectively, and a total of 117 cases were analyzed. The results from recursive partitioning suggested that Pa and performance status could predict the incidence of pneumonia (Figure 3).

## 4. Discussion

We retrospectively analyzed the incidence of pneumonia in patients with RA based on the microorganisms present in the LRT. *P. aeruginosa*, *S. aureus*, and *H. influenzae* were most frequently detected, and *P. aeruginosa* colonization was found to be a probable risk factor for subsequent pneumonia in patients with RA. The annual incidence rates of pneumonia per 1000 patients in these three groups were higher than that in the control group, while the frequency of pneumonia in the control group in this study was similar to that in patients with RA in a previous study [11]. To the best of our knowledge, this is the first study that focuses on microbial colonization of the LRT as a risk factor for pneumonia in patients with RA. The strength of this study lies in the fact that the specimens were collected by bronchoscopy, which may decrease oral bacterial contamination and reflect the LRT-specific flora more accurately than a sputum specimen, although the no-touch technique was not used. Another strength lies in the novel approach used for the evaluation of a possible direct correlation between the presence of *P. aeruginosa* in the LRT and subsequent pneumonia in patients with RA.

*P. aeruginosa* has been reported to be one of the major causative microorganisms of pneumonia in patients with RA [1]. The risks of pneumonia in the Pa group may have been underestimated in this study due to frequent macrolide use in older patients. Long-term macrolide treatment can reduce pulmonary exacerbation and improve bronchiectasis in non-CF patients [12,13]; moreover, macrolides possibly prevent pneumonia in older adults [14]. A relatively higher prevalence of DM and high-degree bronchiectasis in the Pa group may also have affected pneumonia incidences.

Conventional culture methods in non-CF patients with bronchiectasis tend to more frequently under-identify *H. influenzae* than *P. aeruginosa* and *S. aureus* [15]. Therefore, the high incidence of pneumonia in the Hi group reported herein might be due to the small sample size. Further, most cases of pneumonia in the Hi group occurred more than 5 years after colonization, and *P. aeruginosa* was detected in the sputum of two-thirds of the patients who developed pneumonia; thus, microbial substitution by *P. aeruginosa* was possibly related to the incidence of pneumonia in the Hi group.

Reports suggest that *P. aeruginosa* and *H. influenzae* are the most frequent colonizers of the LRT in patients with non-CF bronchiectasis [16,17]. They were also the major colonizers in our study, and they were isolated from cases of bronchiectasis among patients with RA. *S. aureus* colonization with bronchiectasis is associated with CF or allergic bronchopulmonary aspergillosis [17,18], although neither was seen in our patients.

The present study has several limitations. First, it was a retrospective study, and the sample size was small. The incidence of pneumonia may have been overlooked if it was diagnosed or treated at other hospitals. Further, the history of pneumococcal vaccination was unavailable in the medical records; this may have affected the incidence of pneumonia. Second, the detected microorganisms did not reflect findings for the entire LRT. Furthermore, the conventional culture method has limits for detecting bacteria, and anaerobic bacteria may have been underestimated [19]. Third, only one timepoint was assessed, and it is unclear whether microbial substitution occurred after bronchoscopy. To accurately understand the effect of airway microbes on pneumonia in patients with RA, prospective studies with larger samples are required in the future.

## 5. Conclusions

In patients with RA, *P. aeruginosa*, *S. aureus*, and *H. influenzae* were the three major microbes isolated from the LRT, and the incidence of pneumonia was higher in these patients than in the control group. *P. aeruginosa* in the LRT was associated with subsequent pneumonia in patients with RA. Our findings may contribute to the management of patients with RA whose LRT is colonized by *P. aeruginosa*; it is necessary to carefully follow up with these patients, encourage pneumococcal vaccination, and be prepared to treat subsequent pneumonia.

## Figures and Tables

**Figure 1 jcm-10-03552-f001:**
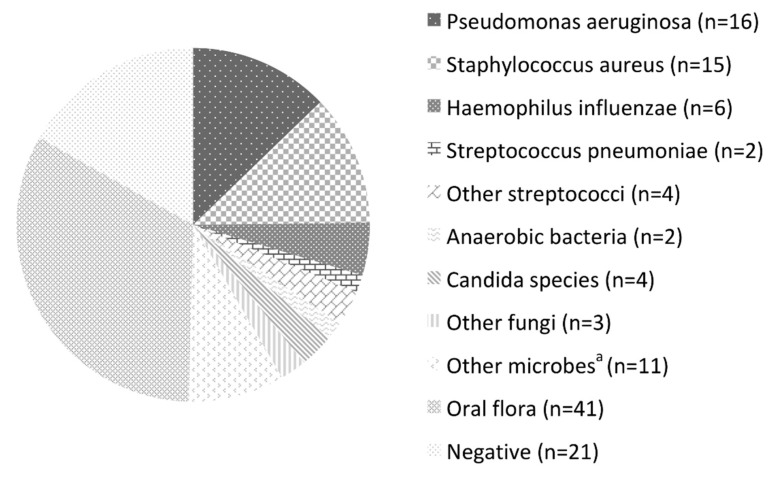
Microorganisms detected by bronchoscopy. ^a^
*Stenotrophomonas maltophilia*, *Burkholderia cepacia*, *Klebsiella oxytoca*, *Pseudomonas fluorescens*, *Serratia marcescens*, *Mycobacterium intracellulare*, and *M. gordonae*.

**Figure 2 jcm-10-03552-f002:**
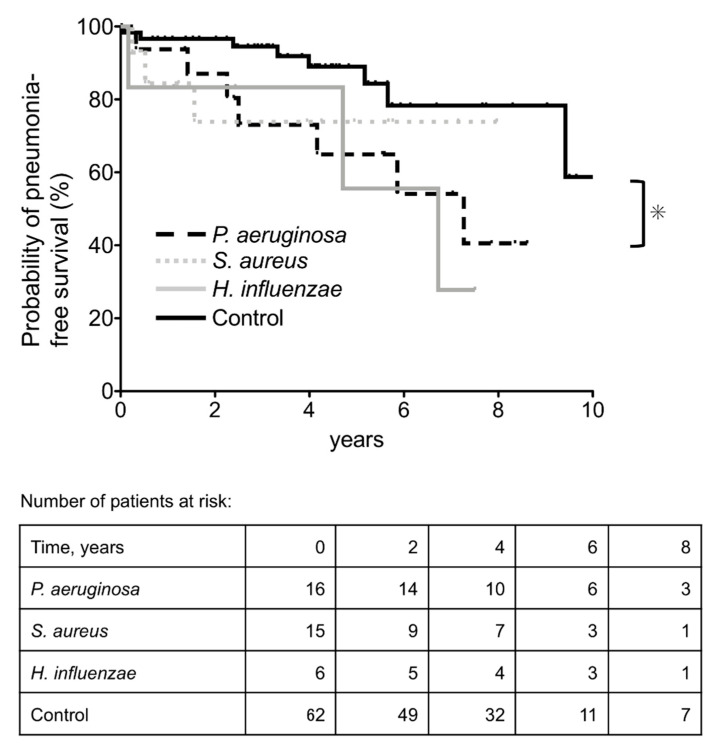
Kaplan–Meier plot and numbers-at-risk table for the probability of pneumonia-free survival. Control group: patients with normal oral flora and those who tested negative for bacteria. * *p* < 0.05.

**Figure 3 jcm-10-03552-f003:**
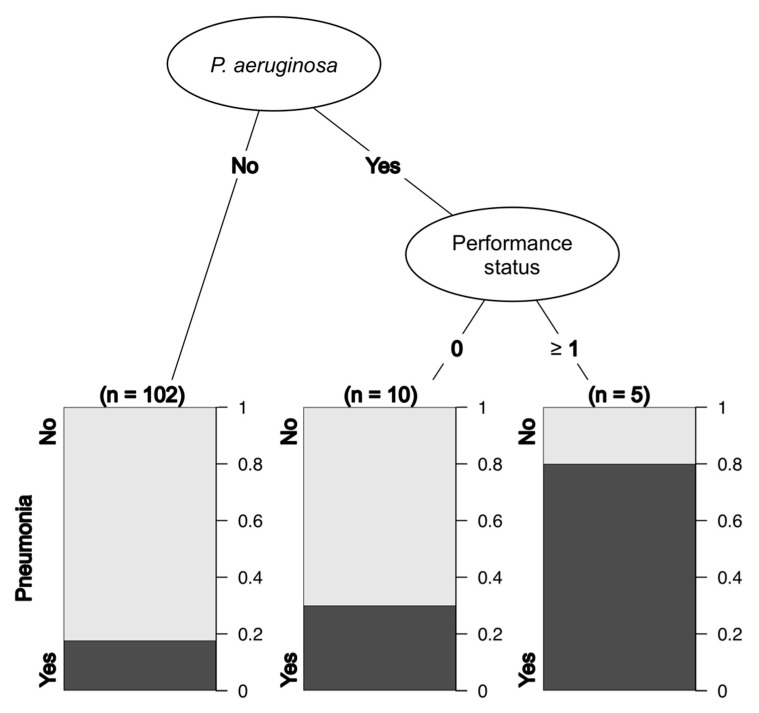
Decision tree analysis for pneumonia.

**Table 1 jcm-10-03552-t001:** Characteristics of patients with rheumatoid arthritis for each microorganism in the lower respiratory tract.

	Total (*n* = 99)	*Pseudomonas aeruginosa* (*n* = 16)	*Staphylococcus aureus* (*n* = 15)	*Haemophilus influenzae* (*n* = 6)	Control (*n* = 62)
Age (years)	67 (61–76)	66 (61–76)	63 (58–76)	69 (64–75)	67 (61–76)
Female patients, %	77.8	87.5	93.3	66.7	72.6
Performance status	1 (0–1)	0 (0–1)	1 (0–1)	1 (0–1)	1 (0–1)
Smoking (pack-years)	0 (0–5)	0 (0–0)	0 (0–0)	0 (0–43)	0 (0–18)
Duration of RA (years) ^¥^	5.2 (0.4–15.6)	3.1 (0.5–8.7)	3.4 (0.1–11.1)	5.8 (1.9–19.7)	6.0 (0.2–16.1)
DAS28-ESR ^§^	4.8 (3.7–6.0)	5.4 (4.1–6.2)	4.6 (3.2–5.7)	4.4 (3.5–4.8)	5.1 (3.6–6.5)
MTX use, %	53.5	62.5	53.3	33.3	53.2
GC use, %	37.4	37.5	6.7	33.3	45.2
Biologics * use, %	23.2	25.0	26.7	16.7	22.6
Macrolide use, %	15.2	43.8	26.7	0	6.5
DM, %	9.1	12.5	6.7	16.7	8.1
ILD, %	34.3	18.8	6.7	33.3	45.2
COPD, %	7.1	12.5	0	16.7	6.5
Modified Reiff score	5 (1–8)	6.5 (4–10.8)	6 (2–12)	1.5 (0.8–7.8)	4 (0.8–6)
Observation period (months)	47 (18–68)	41 (20–86)	45 (6–68)	55 (22–92)	48 (26–66)

Data are presented as frequency (%) or median (interquartile range). RA, rheumatoid arthritis; DM, diabetes mellitus; ILD, interstitial lung disease; COPD, chronic obstructive pulmonary disease; MTX, methotrexate; GC, glucocorticoid; DAS, disease activity score; ESR, erythrocyte sedimentation rate. ^¥^ *n* = 98, ^§^ *n* = 68. * Biologics refer to inhibitors of tumor necrosis factor, interleukin (IL)-1, IL-6, IL-12/23, IL-17, T-cell co-stimulatory molecules, and B-cells.

## Data Availability

Data are contained within the article.

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
