# Peer review of "Infectious Pneumonia and Lower Airway Microorganisms in Patients with Rheumatoid Arthritis"

_jcm, 2021, doi:10.3390/jcm10163552_

Round 1

Reviewer 1 Report

In their retrospective study entitled ‘Infectious pneumonia and lower airway microorganisms in patients with rheumatoid arthritis’ Ideguchi et al. investigate an assumed relationship between rheumatoid arthritis and the occurrence of pneumonia mainly caused by Pseudomonas aeruginosa. In total 121 patients were included in this study, with a confirmed rheumatoid arthritis.
In their study cohort, the authors define 84 subjects as controls, although patients in this group are not only free of bacteria in the respiratory tract or show a normal oral flora, but include also subjects colonized with Streptococcus pneumoniae, Candida, anaerobic and other bacteria and fungi. Patients colonized with two or more pathogens are counted repeatedly, which at first glance leads to confusion and should therefore be explained briefly. I agree that no separate groups should be listed with less than 5 subjects, but assigning these patients at least partially being colonized by lung specific pathogens to the control group does not seem right either. I would therefore suggest listing these patients in table 1 only and excluding from further consideration. Additionally there exists a surplus of female study subjects and I would ask the authors to explain why this is the case. Did I get it right, that smokers occur only in the control and in the Haemophilus groups? And for the Haemophilus group was there one single strong smoker increasing the number of packs up to 43, but due to rest of non-smokers in the group, the mean goes to 0? In the other criteria, the study cohort is well balanced and of sufficient volume. The authors clearly explain the limitations of their study; however, the conclusions drawn are somewhat circular. It would be interesting to compare people with and without RA and the frequency of pneumonia. In the end, the only recommendation of the study is to carefully consider the administration of GC in patients with RA and pneumonia. This might be sufficient for a brief communication, especially as the manuscript is carefully written and the data shown appears valid. The scientific input, however, is to be considered rather low, as there is no information of a possible interplay between RA and pneumonia given at all. 

Minor comments:
I like the scheme displayed in figure 1 showing the enrolment strategy of the study. However, I would assess this as not mandatory, since this is sufficiently described in the text. 
Suggestion: table 1 could be shown as a coloured pie chart which would be more illustrative than a table. 
Please define biologics use in table 2.
Figure 2 defines the control group as ‘patients with microbes other than Pseudomonas aeruginosa, Staphylococcus aureus, and Haemophilus influenza.’ But as far as I understood, the 21 negative subjects are also included here, correct?
Please explain abbreviation DAS-28 for readers not that familiar with RA.

Author Response

Manuscript ID: jcm-1281909 - Minor Revisions

Reviewer: 1

Comments and Suggestions for Authors In their retrospective study entitled ‘Infectious pneumonia and lower airway microorganisms in patients with rheumatoid arthritis’ Ideguchi et al. investigate an assumed relationship between rheumatoid arthritis and the occurrence of pneumonia mainly caused by Pseudomonas aeruginosa. In total 121 patients were included in this study, with a confirmed rheumatoid arthritis.

[C1] In their study cohort, the authors define 84 subjects as controls, although patients in this group are not only free of bacteria in the respiratory tract or show a normal oral flora, but include also subjects colonized with Streptococcus pneumoniae, Candida, anaerobic and other bacteria and fungi. Patients colonized with two or more pathogens are counted repeatedly, which at first glance leads to confusion and should therefore be explained briefly. I agree that no separate groups should be listed with less than 5 subjects, but assigning these patients at least partially being colonized by lung specific pathogens to the control group does not seem right either. I would therefore suggest listing these patients in table 1 only and excluding from further consideration.

(R1) We appreciate your pertinent observations and suggestion. As suggested, we have re-defined the control patients to include only those whose samples tested negative for bacteria, or a normal oral flora, and proceeded to the subsequent analysis.

[C2] Additionally there exists a surplus of female study subjects and I would ask the authors to explain why this is the case.

(R2) We appreciate your pertinent observations and question. Approximately 80% of Japanese patients with RA are female; this has been additionally mentioned on P5, L146-147, and an appropriate citation [10] has been provided for the same.

[C3] Did I get it right, that smokers occur only in the control and in the Haemophilus groups? And for the Haemophilus group was there one single strong smoker increasing the number of packs up to 43, but due to rest of non-smokers in the group, the mean goes to 0?

(R3) We appreciate your pertinent question. There were only two smokers in the Haemophilus group; the result in the table 1 shows the median value.

[C4] In the other criteria, the study cohort is well balanced and of sufficient volume. The authors clearly explain the limitations of their study; however, the conclusions drawn are somewhat circular. It would be interesting to compare people with and without RA and the frequency of pneumonia. In the end, the only recommendation of the study is to carefully consider the administration of GC in patients with RA and pneumonia. This might be sufficient for a brief communication, especially as the manuscript is carefully written and the data shown appears valid. The scientific input, however, is to be considered rather low, as there is no information of a possible interplay between RA and pneumonia given at all.

(R3) We appreciate your pertinent observations, and have revised the conclusions accordingly (P7, L221-223).

Minor comments:

[C5] I like the scheme displayed in figure 1 showing the enrolment strategy of the study. However, I would assess this as not mandatory, since this is sufficiently described in the text.

(R5) We appreciate your observations and have removed figure 1 accordingly.

[C6] Suggestion: table 1 could be shown as a colored pie chart which would be more illustrative than a table.

(R6) We appreciate your observations, and have replaced table 1 by figure 1, which depicts a pie chart.

[C7] Please define biologics use in table 2.

(R7) We appreciate your observations and have added the definitions of biologics in the caption for table 1 (originally table 2).

[C8] Figure 2 defines the control group as ‘patients with microbes other than Pseudomonas aeruginosa, Staphylococcus aureus, and Haemophilus influenza.’ But as far as I understood, the 21 negative subjects are also included here, correct?

(R8) We appreciate your observations and have accordingly re-defined the control group to include those whose samples tested negative for bacteria, or had normal oral flora; the results of re-analysis have been presented in figure 2.

[C9] Please explain abbreviation DAS-28 for readers not that familiar with RA.

(R8) As suggested, the full form for the abbreviation DAS28-ESR has been mentioned on P5, L148.

Reviewer 2 Report

This brief report summarizes original data and is of interest regarding the relationship of microbiological colonization in the LRT and subsequent pneumonia in RA patients.

I have no major concerns. There are a few, mostly methodological suggestions for improvement.

1) My major concern is this one: this is a retrospective study with incidence of pneumonia and probability of pneumonia-free survival being the major endpoint. As no follow up on pneumonia was part of the study protocol, it may be that patients had pneumonia but were treated in another hospital or as outpatients without any records available to the study group leading to an underestimation of pneumonia incidence or an overestimation of pneumonia-free survival. can you please comment on this/make it clearer in the text how this study bias was avoided?

2) it would be interesting to know why patients underwent bronchoscopy (l.105).

3) Table 2: the GC dosage with an average of 0mg/day and duration of GC use of 0 months probably refers to all patients/all patients of the respective group. I would suggest to restrict GC dosage and duration of GC use to only patients who received any GC to see if there are differences in the groups. Otherwise this information is of no use.

5) l. 171 the authors state that oral bacterial contamination was minimized. This probably refers to other sampling methods like sputum sampling, but as no no-touch bronchoscopy technique was used, oral bacterial contamination of the bronchoscope is still likely. Please adapt.

Author Response

Reviewer: 2

Comments and Suggestions for Authors

This brief report summarizes original data and is of interest regarding the relationship of microbiological colonization in the LRT and subsequent pneumonia in RA patients.

I have no major concerns. There are a few, mostly methodological suggestions for improvement.

[C1] My major concern is this one: this is a retrospective study with incidence of pneumonia and probability of pneumonia-free survival being the major endpoint. As no follow up on pneumonia was part of the study protocol, it may be that patients had pneumonia but were treated in another hospital or as outpatients without any records available to the study group leading to an underestimation of pneumonia incidence or an overestimation of pneumonia-free survival. can you please comment on this/make it clearer in the text how this study bias was avoided?

(R1) We appreciate your pertinent observations. As correctly observed, pneumonias diagnosed and treated at other hospitals could have been overlooked in this study. We have therefore additionally mentioned this in the paragraph on limitations in the discussion section (P7, L204-205). Data regarding pneumonia diagnosed by participating hospitals in the study was obtained from the electronic medical records; this included antimicrobial use history (P3, L96-97).

[C2] it would be interesting to know why patients underwent bronchoscopy (l.105).

(R2) We appreciate your pertinent observations; the reasons for performing bronchoscopy have been added to the results section for clarity (P3, L113-115).

[C3] Table 2: the GC dosage with an average of 0mg/day and duration of GC use of 0 months probably refers to all patients/all patients of the respective group. I would suggest to restrict GC dosage and duration of GC use to only patients who received any GC to see if there are differences in the groups. Otherwise this information is of no use.

(R3) We appreciate your pertinent observations and suggestion. On restricting the analysis only to patients with GC use, there was no difference between the groups on comparing the GC dosage and duration of GC use. We have therefore removed these variables from table 1.

[C4] l. 171 the authors state that oral bacterial contamination was minimized. This probably refers to other sampling methods like sputum sampling, but as no no-touch bronchoscopy technique was used, oral bacterial contamination of the bronchoscope is still likely. Please adapt.

(R4) We appreciate your pertinent observations. The text in the discussion section has been revised accordingly (P6, L178).